# Phytochemical profiling and GC-MS analysis of bioactive compounds in methanolic crude extract of *Beta vulgaris* (BV) root from Bangladesh

**Sarder Arifuzzaman**⬥*, **Zubair Khalid Labu, Md. Harun-Or-Rashid, Md. Mehedi Hasan, Abdullah Al Maruf, Sushanto Tappo, Farhina Rahman Laboni**

Department of Pharmacy, World University of Bangladesh, Uttara, Dhaka, Bangladesh

* sarder.arifuzzaman@pharmacy.wub.edu.bd, arifpharmju@gmail.com

## Abstract

*Beta vulgaris* (BV) is distributed worldwide and has long been used as a culinary and in traditional medicine to treat diseases. The objective of the present study was to investigate the phytochemical constituents and the chemical fingerprinting of *BV* root of Bangladeshi origin. We performed qualitative conventional lab tests of colorimetric reactions with specific reagents to identify the presence of various phytochemical classes like alkaloids, flavonoids, saponins, carbohydrates, reducing sugars, tannins and steroids. To gain comprehensive insights into the chemical composition, we used gas chromatography and mass spectroscopy analysis (GC-MS). Finally, we employed computational methodologies, leveraging *in-silico* physicochemical properties, pharmacokinetics, drug-likeness and medicinal chemistry friendliness analysis to identify compounds with favorable properties, increasing the likelihood of successful drug development. Phytochemical screening indicated that methanolic extract is rich in alkaloids, tannins, flavonoids, saponins, triterpenes, glycosides and carbohydrates. GC-MS analysis revealed the presence of 69 chemicals, including alkaloids and amines, amino acids and derivatives, esters, and carbohydates. The annotation of the biological function of these compounds revealed many of them or their derivatives have reported disease-modifying functions (e.g., antidiabetic, antioxidant, anti-inflammatory, anticancer, cardioprotective, etc.). Further, cheminformatics analysis revealed that at least 20 molecules (e.g., 5-(hydroxymethyl)-2-Pyrrolidinone, Pidolic acid, etc) possess not only higher concentration in beetroot but also a favorable profile for drug development. Overall, our findings of the present study contribute to understanding that the *BV* root can be used as a valuable source in the field of natural products drug discovery.

which permits unrestricted use, distribution, and reproduction in any medium, provided the original author and source are credited.

**Data availability statement:** All data are in the manuscript and/or Supporting Information files.

**Funding:** The author(s) received no specific funding for this work.

**Competing interests:** The authors declare that they have no competing interests.

## Author summary

We conducted an in-depth phytochemical and computational analysis of the methanolic crude extract of *Beta vulgaris* (BV) root cultivated in Bangladesh. Preliminary phytochemical screening confirmed the presence of phenolic and flavonoid constituents, and we quantitatively evaluated them through total phenolic content (TPC) and total flavonoid content (TFC). Gas Chromatography–Mass Spectrometry (GC–MS) analysis identified a broad range of bioactive compounds. Systematic classification based on chemical nature of these compounds grouped them into phenolics, organic acids, esters, alcohols and alkaloids. Many of these compounds reported to exhibit antioxidant, antimicrobial, anti-inflammatory, antidiabetic and anticancer properties, which suggest the potential therapeutic benefits of BV root extract. We further examined selected compounds for their structural characteristics, molecular properties, *in-silico* pharmacokinetics. Drug-likeness and ADME (absorption, distribution, metabolism and excretion) predictions revealed promising pharmacological potential, with several compounds satisfying essential drug development criteria. By integrating phytochemical profiling with computational drug discovery approaches, our study not only highlights *Beta vulgaris* root as a valuable natural reservoir of therapeutic molecules but also strengthens the evidence for the medicinal value of beetroot and underscores its potential applications in modern pharmacology.

## Introduction

Beet (*Beta vulgaris*) is cultivated worldwide for its edible leaves and roots (Fig 1). Roots are frequently roasted or boiled and served as a side dish. They are also commonly canned, either whole or cut up, and often are pickled, spiced, or served in a sweet-and-sour sauce [1]. It is known that beetroots are a good source of riboflavin as well as folate, manganese, iron, and vitamins A, C, K and the antioxidant betaine [2]. Betanin, obtained from the root, is used industrially as red food colorant to enhance the color and flavor of tomato paste, sauces, desserts, jams and jellies, ice cream, candy, and breakfast cereals [3]. There are clinical trials reported that consumption of beetroot juice modestly reduced systolic blood pressure but not diastolic blood pressure [4,5].

Beetroot extracts, rich in bioactive compounds like flavonoids, inhibited growth-related signaling pathways and reduced apoptotic and cell cycle proteins levels, indicating their potential in cancer treatment [6]. Many studies have investigated and documented the chemical composition of beetroot, revealing a rich source of various nutrients and bioactive compounds [7–10]. However, to date, no study has been surfaced addressing the phytochemical profiling of the beetroot grown in Bangladesh. Thus, there is a strong rationale to validate the ethnomedicinal uses with identification and quantification of chemical compounds of beetroot extract of Bangladesh origin.

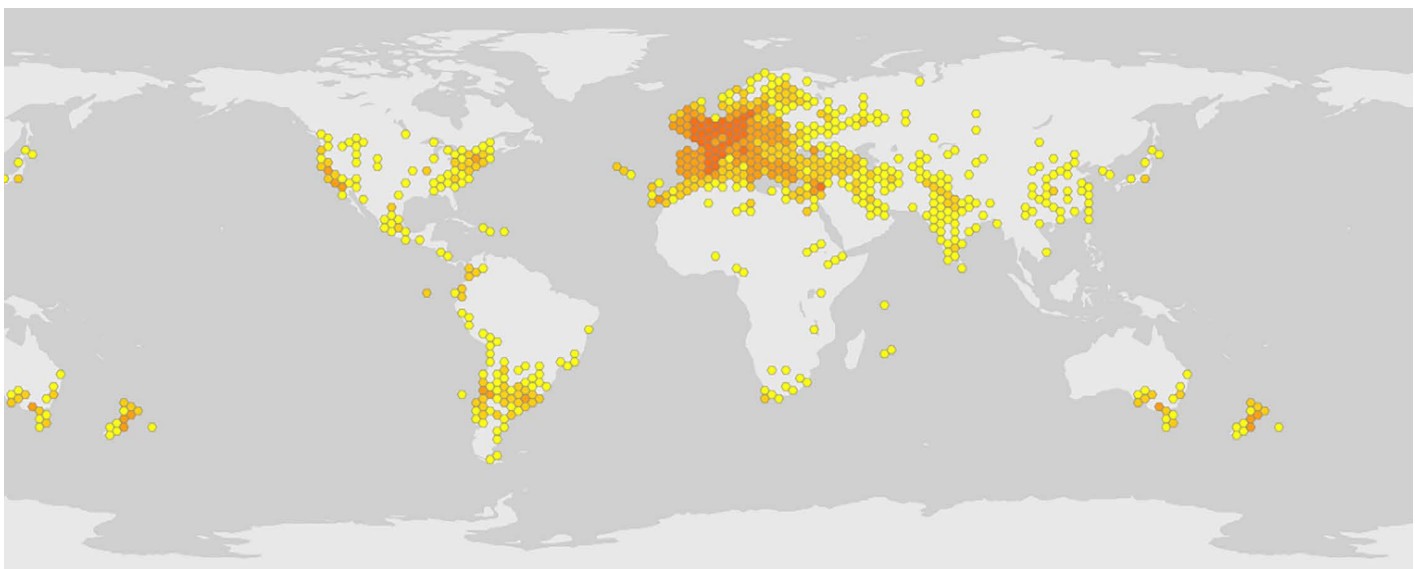

**Fig 1. Distribution of *Beta vulgaris* plant.** The classical color (yellow and orange) hexagonal symbol indicates the distribution of BV on the world map. Data curated from the Global Biodiversity Information Facility (https://www.gbif.org/) for illustrative purposes only.

There are several conventional laboratory tests used for phytochemical screening to detect various groups of phytochemicals [11]. These include Wagner's and Dragendorff test for alkaloid, Shinoda test for flavonoid, Ferric chloride test for phenol, Molisch's test for Carbohydrate, Fehling solution test for reducing sugar, Libermann-Burchard test for steroid, Lead acetate test for tannin, etc. [11]. Qualitative assays for phytochemical classes (alkaloids, flavonoids, saponins, tannins, terpenoids, etc.) provide rapid insights into extract composition. They are simple, however, they are generally qualitative, providing information about the presence or absence of specific compounds [11]. Methods for identifying such medicinal compounds requires modern, simple and repeatable.

One of the modern methods for identifying these compounds is gas chromatography–mass spectrometry (GC–MS), which can isolate and analyze compounds in a single step using a mass detector and available GC–MS libraries [11]. GC–MS enables identification of volatile/semi-volatile metabolites (fatty acids, amines, esters, etc.), and is a crucial technique for analyzing plant extracts, offering a fast and accurate method for identifying compounds, including active principles [12]. The application of GC-MS allows researchers to quantify the presence of specific bioactive compounds in herbs, providing valuable insights into their potential medicinal properties [12]. Reported GC–MS studies of beetroot (mostly ethanol/methanol extracts) have identified compounds such as dimethyl alkylamines, fatty acid methyl esters, long-chain ketones, phytols and others [7,9,10,13]. However, these profiles vary by cultivar, extraction and geography.

Reports described that climate variations, including temperature, rainfall, and UV radiation, directly influence the concentration and composition of bioactive compounds in plants [8,13]. These environmental factors can significantly alter the levels of beneficial chemicals produced by plants, leading to variations in the potency and type of bioactive compounds found in a species across different geographical locations. This is because plants adapt to their environment by adjusting their chemical profiles in response to environmental stressors like extreme weather [13,14]. In silico cheminformatics (e.g., SwissADME) offers rapid prediction of physicochemical properties, pharmacokinetics and drug-likeness of plant compounds. For instance, beetroot flavonoids and betalains have been evaluated for Lipinski's rule, solubility and absorption to gauge their nutraceutical potential [15]. This study therefore aimed to profile the methanolic crude extract of *B. vulgaris* root from Bangladesh by: (1) qualitative phytochemical screening for major metabolite classes, (2) GC–MS analysis to

identify and quantify major constituents, and (3) in silico ADME/drug-likeness analysis of key compounds. The results are compared to published data, and their implications for functional-food and nutraceutical uses (culinotherapeutics) are discussed.

## Materials and methods

### Chemicals and instruments

Analytical-grade methanol, ethanol, chloroform, n-hexane, ethyl acetate, Chloroform, Dragendorff's reagent, wagner's reagent, ferric chloride, lead acetate, concentrated sulfuric acid ($H_2SO_4$) and hydrochloric acid (HCl) were procured from Merck (Darmstadt, Germany). All solvents used for extraction and GC-MS analysis were GC-grade. Whatman No.1 filter papers (GE Healthcare, UK) were used for filtration. Phytochemical tests were carried out using standard laboratory glassware and reagent-grade chemicals. The rotary evaporator (Heidolph, Germany) was used for solvent removal under reduced pressure. GC-MS profiling of the methanolic crude extract was performed on Shimadzu GC-MS-QP2010 (Japan). Before initiation of the experiment, all chemicals were stored according to the indicated storage guidelines of each chemical. Trained personnel have operated all instruments.

### Collection and identification of plant samples

The fresh plant was collected from Dinajpur district, Bangladesh. Freshly collected BV plants were taken to the Bangladesh National Herbarium in Mirpur, Dhaka, for authentication. The voucher specimen was added to the Bangladesh National Herbarium and given the accession number 112753.

### Drying and grinding of plant materials

The collected roots were cleaned under clean running water to remove any remaining dirt. The samples were first allowed to dry at room temperature under the shade for a week, then dried at 50–60°C in a mechanical dryer to achieve complete drying. The dried roots were mechanically ground into a coarse powder. The powdered sample was stored in a sealed, airtight container in a cool, dry, and dark place till further use.

### Extract preparation

Four hundred grams (100 g) of root powder was weighed and dissolved in 500 mL of methanol and extracted using a Soxhlet extraction apparatus for 72 h. The methanol solvent was evaporated from the extraction by using a rotary evaporator under reduced pressure to obtain the methanol crude extract. A redish gooey and sticky concentrated methanol crude extract was used for phytochemical screening and pharmacological activity evaluation. The extract was stored at 40ºC in a Pharmaceutical standard refrigerator until further use. Twenty-five grams of crude extract were dissolved and extracted with methanol, ethanol, Acetone, Ethyl acetate, n-hexane, and chloroform. The dried extractive amounts were methanol (5.0g), ethanol (1.5g), Acetone (1.5g), Ethyl acetate (0.5g), n-hexane (1.2g) and Chloroform (1.02g). All crude extracts were filtered separately through Whatman No. 41 filter paper to remove particles. The particle-free crude extract was evaporated completely by using rotary evaporator under reduced pressure to obtain dry crude extracts. The residue left in the separatory funnel was re-extracted twice following the same procedure and filtered. The combined extracts were concentrated and dried by using a rotary evaporator under reduced pressure.

### Phytochemical screenings

All crude extracts were subjected to preliminary phytochemical screening to detect the presence of major bioactive constituents, following standard protocols with slight modifications [16,17]. All extracts were dissolved in distilled water (DW) to a final concentration of 1 mg/mL. Alkaloids were analyzed using Wagner's and Dragendorff's qualitative tests. In Wagner's

test, 0.5 mL of extract was mixed with 1 mL of 1% (v/v) HCl, followed by the addition of three drops of Wagner's reagent; the appearance of brown or reddish precipitates indicated a positive result. In Dragendorff's test, we added 1 mL of Dragendorff's reagent to 2 mL of plant extract in a test tube and observed an orange-red mixture to confirm the presence of alkaloids. We purchased caffeine, Wagner's reagent, and Dragendorff's reagent from Sigma-Aldrich(USA).

Flavonoids were detected using the alkaline reagent assay. Briefly, 2–3 drops of sodium hydroxide (NaOH) were added to 2 mL of plant extract, followed by 5 mL of dilute hydrochloric acid (HCl). The formation of a yellow coloration that became colorless after acidification indicated a positive result. Quercetin was used as a standard control. We obtained NaOH and quercetin from Merck (Germany). Polyphenols were detected by adding 2 mL of Folin-Ciocalteu's reagent to 0.5 mL of extract, followed by the addition of 2.5 mL of 7.5% $Na_2CO_3$ solution. Development of a blue coloration indicated the presence of polyphenols. Gallic acid was employed as a standard control. Folin-Ciocalteu's reagent and gallic acid were purchased from Merck (Germany). Tannins were detected by adding three drops of 1% (w/w) ferric chloride ($FeCl_3$) to 1 mL of plant extract. The formation of a blue-black or green-black coloration indicated a positive result. We purchased $FeCl_3$ and gallic acid from Merck (Germany).

For steroid detection, 2 mL of acetic anhydride and 2 mL of concentrated sulfuric acid ($H_2SO_4$) were added to 5 mL of aqueous plant extract. A color change from violet to blue-green indicated the presence of phytosterols. Terpenoids were assessed by adding 2 mL of chloroform and 1–2 drops of concentrated $H_2SO_4$ to 5 mL of beetroot extract, shaking the mixture and allowing it to stand; formation of a reddish-brown interface confirmed the presence of terpenoids. For carbohydrate detection, 2–3 drops of Molisch's reagent were added to 2 mL of beetroot extract, followed by gentle addition of 1 mL of concentrated $H_2SO_4$ along the wall of the test tube. The formation of a purple-colored ring at the interface indicated the presence of carbohydrates. Molisch's reagent and glucose (standard control) were purchased from Merck (Germany). All assays were performed in triplicate with appropriate positive (standard compounds) and negative (distilled water) controls to ensure accuracy and reproducibility.

## Estimation of total phenolic content (TPC)

We estimated the total phenolic content (TPC) using the Folin–Ciocalteu reagent method with gallic acid as the standard, following a protocol based on Arifuzzaman S. et al. (2025) with slight modifications [18]. Both Folin & Ciocalteu's phenol reagent and gallic acid were purchased from Merck, Germany. Briefly, 0.5 mL of a different solvent extract (1 mg/ml) was combined with 2 mL of Folin & Ciocalteu's phenol reagent, which had been previously diluted tenfold with deionized water. After mixing and allowing the mixture to react for 5 minutes at room temperature ($25^0C$), 2.5 mL of 7.5% sodium carbonate ($Na_2CO_3$) was added to the mixture and gently stirred. $Na_2CO_3$ solution neutralizes the reaction and promotes color development. The reaction mixture was then incubated in the dark at $25 \pm 2°C$ for 30 minutes to ensure complete color formation.. The absorbance of the blue-colored complex was then measured at 760 nm using a UV spectrophotometer (Model: UV-1700 series). A standard calibration curve was generated using gallic acid solutions ranging from 10 to 60 mg/ml, with the resulting equation $y = 0.0086x - 0.1111$ ($R^2 = 0.9954$). The TFC was calculated from the calibration curve and expressed as milligrams of gallic acid equivalents per gram of dry weight (mg GAE/g DW) of the sample. All samples and standards were analyzed in triplicate to ensure reproducibility, and the results were reported as mean $\pm$ standard deviation (SD).

## Estimation of total flavonoid content (TFC)

We also estimated TFC using the aluminum chloride colorimetric method with quercetin as the standard, following a modified procedure adapted from previous studies [18]. Briefly, 1.5 ml of the different solvent extracts (1 mg/ml) was mixed with 0.1 ml of 10% (w/v) aluminum chloride ($AlCl_3$) solution and 0.1 ml of 1 M sodium acetate ($CH_3COONa$). The reaction mixture was then incubated at room temperature ($25 \pm 2°C$) for 30 minutes to allow for complex formation., Then, we added

1 mL of 1M NaOH solution and adjusted the final volume of the mixture to 5 mL using double-distilled water.. After allowing the final mixture to stand for an additional 15 minutes at room temperature ($25 \pm 2°C$), the absorbance of the solution was measured at 415 nm using a UV spectrophotometer (Model: UV-1700 series).

A calibration curve was prepared using quercetin standard solutions ranging from 10 to 100 µg/ml. TFC was calculated from the calibration curve and expressed as milligrams of quercetin equivalents per gram of dry weight (µg QE/g DW) of the sample. The yielding the equation $y = 0.01x - 0.0409$ with a high degree of linearity ($R^2 = 0.9921$). We purchased AlCl3, $CH_3COONa$, NaOH, and quercetin from Merck, Germany. All measurements were performed in triplicate and results were reported as mean ± standard deviation (SD).

## Gas chromatography–mass spectroscopy (GC–MS) analysis

GC-MS profiling of the methanolic crude extract was performed on a Shimadzu GC-MS-QP2010 Plus (Japan) system equipped with an Rtx-5MS fused silica capillary column (30 m × 0.25 mm i.d., 0.25 µm film thickness), an auto sampler, and an electron ionization detector. This column is suitable for analyzing a wide range of samples, including plant extracts and other complex matrices. For analysis, 1 mg/mL of the crude extract was prepared in GC-grade methanol and injected (1.0 µL) into the GC Machine. The GC's parameters were set up as follows: auxiliary temperature: 280°C; carrier gas (Helium) flow rate: 1.1 ml/min; inlet temperature: 250°C; oven temperature: 90°C at 0 min, increased to 200°C for 2 min (3°C/min), then 280°C for 2 min (15°C/min). The total retention duration of the chromatographic analysis was 50 minutes. The following were the MS parameters that were set: quad temperature: 150°C; source temperature: 230°C; mass range: 50–650 m/z; mode: scan mode. Data acquisition and spectral matching were carried out using Shimadzu GC-MS Solution software integrated with NIST-MS Library (Ver. 3.4.5) and Wiley mass spectral libraries. The relative percentage of separated compounds was determined from the peak areas of the total ionic chromatogram (TIC).

## Chemical structure drawing, visualization and chemoinformatics analysis

We used ChemAxon's MarvinSketch software (MarvinSketch and Calculator Plugins, version 23.3.0), a comprehensive cheminformatics tool for molecular drawing, structure optimization and property prediction [19]. The chemical structures of the compounds were drawn manually by pasting the IUPAC names identified by GC-MS analysis. After verifying the structural integrity, the molecules were subjected to 2D clean-up to ensure optimal geometry. MarvinSketch's integrated calculators were employed to determine key physicochemical descriptors, including molecular weight (MW), partition coefficient (logP), topological polar surface area (TPSA), hydrogen bond donors (HBD), hydrogen bond acceptors (HBA) and rotatable bonds.

## *In-silico* pharmacokinetic parameter analysis

In in silico models, PK parameters are predicted using quantitative structure–activity relationships (QSAR), machine learning models, or empirical formulas based on molecular descriptors. We used SwissADME (online platform), a free web-based tool developed by the Swiss Institute of Bioinformatics (SIB). It predicts a wide range of physicochemical properties, pharmacokinetics, drug-likeness, and medicinal chemistry friendliness of small molecules based on their chemical structure [20]. It is widely used in early-stage drug discovery to quickly assess compounds without needing experiments [20]. Compound structures were drawn in ChemAxon MarvinSketch and exported in SMILES, then uploaded to SwissADME for analysis.

The pharmacokinetic properties of the bioactive compounds were predicted using the SwissADME online platform, which evaluates absorption, distribution, metabolism and excretion (ADME) characteristics based on chemical structure. P-glycoprotein (P-gp) substrate status is predicted using support vector machine (SVM) classification based on molecular fingerprints and topological descriptors, providing a binary output (substrate/non-substrate). Cytochrome P450 (CYP450) enzyme inhibition for major isoforms (CYP1A2, CYP2C9, CYP2C19, CYP2D6, CYP3A4) is predicted using random forest

models trained on known inhibitor datasets. All predicted ADME properties, including GI absorption, BBB permeability, P-gp substrate status, and CYP450 inhibition profiles, were recorded for each compound and used to assess drug-likeness, pharmacokinetic behavior and therapeutic potential.

## Statistical analysis

Results were expressed as mean±SEM. To determine the statistical significance one way ANOVA followed by Dunnett's multiple comparisons was performed.

## Results

### Phytochemical screening of beetroot extracts prepared using different solvents

First, we performed qualitative phytochemical screening tests on beetroot extracts prepared with seven different solvents—methanol, ethanol, acetone, ethyl acetate, n-hexane, chloroform, and distilled water—using standard analytical assays. Table 1 summarizes the results. We detected alkaloids, flavonoids, polyphenols, and carbohydrates in all solvent extracts, including the aqueous extract, as confirmed by positive results in Wagner's and Dragendorff's tests, alkaline reagent assay, Folin-Ciocalteu assay and Molisch's test, respectively (Table 1). This indicates that these phytochemical groups are widely distributed and can be extracted using solvents of varying polarity.

However, we did not find tannins, steroids or terpenoids in any of the organic solvent extracts. Instead, we detected these compounds only in the distilled water extract, as shown by positive results in the lead acetate test (tannins) and Salkowski test (terpenoids) (Table 1). These findings suggest that beetroot tannin and terpenoid phytochemicals are water-soluble or better extracted with aqueous solvents. Our findings highlight that alkaloids, flavonoids, polyphenols, and carbohydrates are broadly extractable across polar and non-polar solvents, while tannins and terpenoids require aqueous extraction, emphasizing the critical role of solvent choice in phytochemical extraction from beetroot (Table 1).

### Estimation of total phenolic contents (TPC) and total flavonoid content (TFC)

Total Phenolic Content (TPC) and Total Flavonoid Content (TFC) of beetroot extracts were quantified in various solvents with differing polarities (Fig 2 and Table 2). The solvents are arranged in order of increasing polarity, ranging from non-polar (hexane and chloroform) to highly polar (distilled water). Our results indicate a clear trend in both TPC and TFC values with respect to solvent polarity. Non-polar solvents such as hexane and chloroform yielded the lowest TPC (5.4±0.2 and 8.7±0.3 mg GAE/g extract, respectively) and TFC (3.1±0.1 and 4.5±0.2 µg QE/g extract, respectively). As solvent polarity increased, both phenolic and flavonoid extraction efficiencies improved markedly. Ethyl acetate, a moderately

**Table 1. Phytochemical screening of beet root extracts in different solvents.**

| Phytochemicals | Test Method | Methanol | Ethanol | Acetone | Ethyl acetate | n-hexane | Chloroform | Distilled Water |
|---|---|---|---|---|---|---|---|---|
| Alkaloids | Wagner's and Dragendorff test | + | + | + | + | + | + | + |
| Flavonoids | Alkaline reagent assay | + | + | + | + | + | + | + |
| Polyphenols | Folin-Ciocalteu test | + | + | + | + | + | + | + |
| Tannins | Lead acetate test | + | – | – | – | – | – | + |
| Steroids | Liebermann-Burchard's test | – | – | – | – | – | – | + |
| Terpenoids | Salkowski Test | – | – | – | – | – | – | + |
| Carbohydrates | Molisch's Test | + | + | – | + | – | – | + |

Each phytochemical group is associated with a recognized test method and the result is recorded as either positive (+) or negative (–), indicating the presence or absence of the phytocompounds, respectively.

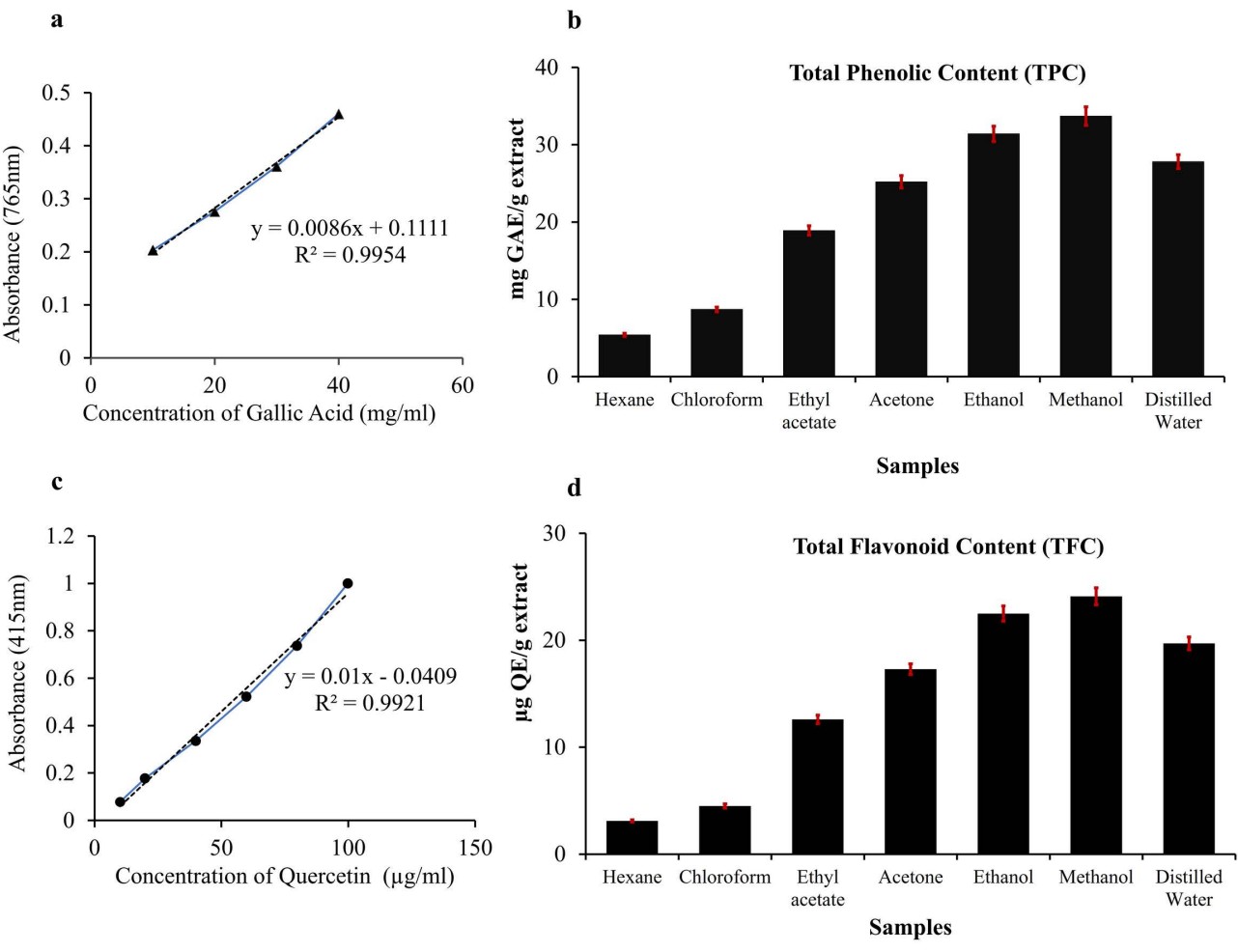

**Fig 2. Estimation of Total Phenolic and Flavonoid Content of Beetroot Powder Extracted with different Solvents (n = 3): (a) Standard curve of Gallic acid for the quantification of Total phenolic content (TPC), (b) Concentration of TPC (mg GAE/ g dry extract); (c) Standard curve of Quercetin for the quantification of Total Flavonoid Content (TFC), and (d) Concentration of TFC (µg QE/ g dry extract) done in triplicate.**

polar solvent, extracted 18.9 ± 0.6 mg GAE/g TPC and 12.6 ± 0.4 µg QE/g TFC. Polar aprotic acetone showed further increased values (25.2 ± 0.8 mg GAE/g and 17.3 ± 0.5 µg QE/g for TPC and TFC, respectively).

Among all solvents tested, methanol, a polar protic solvent, achieved the highest extraction efficiency, with TPC and TFC values of 33.7 ± 1.2 mg GAE/g and 24.1 ± 0.8 µg QE/g, respectively, indicating it is the most effective solvent among those tested for extracting phenolics and flavonoids from beetroot, which agreed with results of Edziri et al. (2022), being 39.75 mg/g [21]. Ethanol and distilled water, both polar solvents, also demonstrated high extraction capabilities, though slightly lower than methanol. These findings suggest that solvent polarity significantly influences the extraction of phenolic and flavonoid compounds from beetroot, with polar protic solvents such as methanol and ethanol being most effective. This underscores the importance of solvent selection in maximizing the recovery of bioactive compounds in phytochemical studies.

### Identification and quantification of chemical compounds in the methanolic extract

GC-MS analysis identified and quantified 69 compounds in the methanolic extract of beetroot. The GC-MS chromatogram is presented in Fig 3, while the chemical compounds with their retention time (RT), molecular formula, molecular weight

**Table 2. Total phenolic content (TPC) and Total flavonoid content (TFC) in beetroot of different solvent extracts.**

| Solvent | Polarity | Total Phenolic Content (TPC) (mg GAE/g extract) | Total Flavonoid Content (TFC) (µg QE/g extract) |
|---|---|---|---|
| Hexane | Non-polar | 5.4±0.2 | 3.1±0.1 |
| Chloroform | Non-polar | 8.7±0.3 | 4.5±0.2 |
| Ethyl acetate | Moderately polar | 18.9±0.6 | 12.6±0.4 |
| Acetone | Polar aprotic | 25.2±0.8 | 17.3±0.5 |
| Ethanol | Polar protic | 31.4±1.0 | 22.5±0.7 |
| Methanol | Polar protic | 33.7±1.2 | 24.1±0.8 |
| Distilled Water | Highly polar | 27.8±0.9 | 19.7±0.6 |

Notes: TPC expressed as milligrams of gallic acid equivalents (GAE) per gram of extract, TFC expressed as milligrams of quercetin equivalents (QE) per gram of extract. Values are presented as mean±standard deviation (n=3). Solvents are listed in approximate order of increasing polarity.

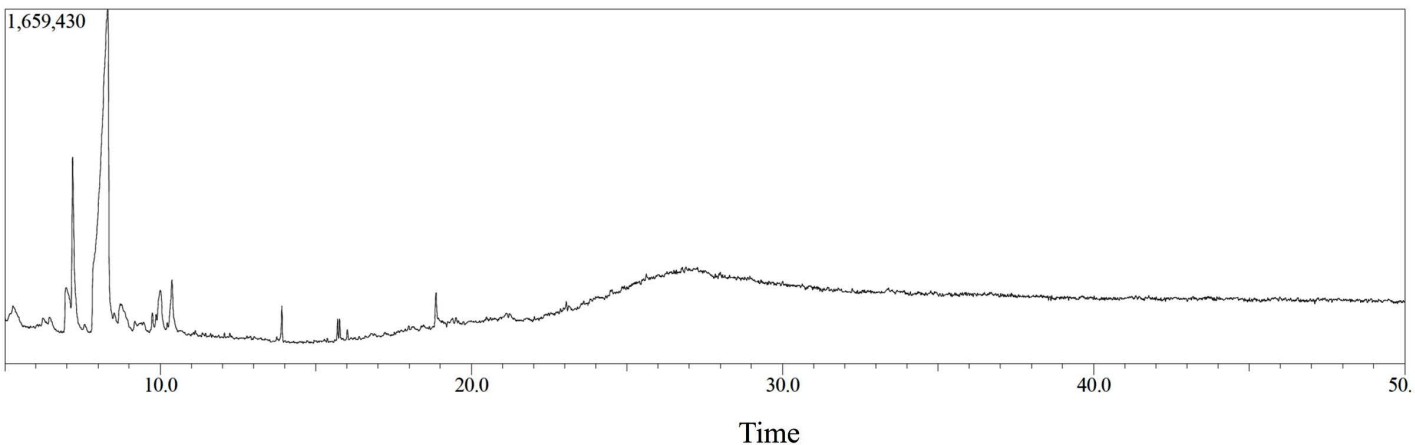

**Fig 3. GC-MS chromatogram for the methanolic extract of beetroot.**

(MW), and concentration (%) in the methanol extract are presented in Table 3. The retention times range from around 5.4 minutes to 18.9 minutes. The variation in retention times is indicative of the differences in the chemical properties of the compounds (e.g., polarity, molecular size, and interactions with the stationary phase in the chromatography column). Compounds with lower retention times (around 5–7 minutes) are likely smaller, less polar, or interact less with the stationary phase of the chromatography column, allowing them to elute faster. Compounds with intermediate retention times (around 8–12 minutes) have moderate polarity, size, or a balance of interactions with the stationary phase. While compounds with higher retention times (around 13–18 minutes) are likely larger or more polar, which results in stronger interactions with the stationary phase, causing them to elute more slowly.

The compounds listed in Table 3, include concentrations ranging from 0.048 µg/mL to over 6 µg/mL. The concentration values appear to vary widely, indicating the presence of different substances in various amounts. Several compounds are present in relatively high concentrations of 6.416 µg/mL, including L-Glutamine, DL-Proline, 5-oxo-, methyl ester, and 2-Pyrrolidinone, 5-(hydroxymethyl)-. These compounds likely make up a significant portion of the sample's composition. Compounds such as 1,3-Propanediol, 2-ethyl-2-(hydroxymethyl)-, Sucrose, and Cyclopentanol are at 3.314 µg/mL, indicating a moderate presence in the sample. Several compounds, including

**Table 3. List of compounds identified and quantified by GC-MS analysis of the methanolic extract of beetroot. Classification of the compounds based on functional group and structural feature. Potential therapeutic use of the compounds curated by the knowledgebase Drug-Bank (https://go.drugbank.com/) and previously published articles.**

| ID# | Name | Pubchem CID | Chemical Formula | Retention Time | m/z | Area | Conc. (ug/mL) | Chemical Class | Possible therapeutic use | Ref. |
|---|---|---|---|---|---|---|---|---|---|---|
| 1 | 2-Heptanamine, 5-methyl- | 541442 | C8H19N | 5.426 | 44.00 | 16716 | 0.070 | Alkaloid (Amine) | Anti-leishmanial, acts as a CNS stimulant; could improve alertness, focus, and energy. | [22] |
| 2 | Propanenitrile, 3-amino-2,3-di(hydroxymino)- | 136259682 | C3H4N4O2 | 5.426 | 44.00 | 16716 | 0.070 | Nitrile derivative | Possible antioxidant, psychoactive and anti-cancer research (chelates metals, reducing oxidative stress). | [23] |
| 3 | Hexanal | 6184 | C6H12O | 5.426 | 44.00 | 16716 | 0.070 | Aldehyde | Exhibits antimicrobial, antifungal and anti-inflammatory potential; investigated for cancer inhibition. | [24,25] |
| 4 | 3-Azabicyclo[3.2.2] nonane | 9240 | C8H15N | 5.426 | 44.00 | 16716 | 0.070 | Bicyclic Amine | Antiprotozoal, scaffold for drugs affecting the nervous system (e.g., potential memory enhancers). | [26] |
| 5 | 2-Pyrrolidinone, 5-(hydroxymethyl)- | 558359 | C5H9NO2 | 7.181 | 84.00 | 1527768 | 6.416 | Lactam (cyclic amide) | Potential neuroprotective agent; may reduce brain cell damage. | [27,28] |
| 6 | DL-Proline, 5-oxo-, methyl ester | 78646 | C6H9NO3 | 7.181 | 84.00 | 1527768 | 6.416 | Amino acid derivative | Building block for peptides with anti-inflammatory, antidiabetic, antioxidant, and wound-healing properties. | [29] |
| 7 | L-Glutamine | 5961 | C5H10N2O3 | 7.181 | 84.00 | 1527768 | 6.416 | Amino Acid | Antidiabetic, cardioprotective, speeds up healing, improves gut barrier function, reduces muscle wasting in illness. | [30–32] |
| 8 | 2-Piperidinemethanol | 94263 | C6H13NO | 7.181 | 84.00 | 1527768 | 6.416 | Amine Alcohol | Basis for developing antidepressants and antipsychotics. | |
| 9 | Propane, 2-isocyanato-2-methyl- | 62412 | C5H9NO | 7.181 | 84.00 | 1527768 | 6.416 | Isocyanate | No therapeutic benefit | |
| 10 | Pidolic acid | 7405 | C5H7NO3 | 7.181 | 84.00 | 1527768 | 6.416 | Pyroglutamic acid | Enhances memory, reduces cognitive decline; used in brain supplements. | [33,34] |
| 11 | 2-Ethylpiperidine | 94205 | C7H15N | 7.181 | 84.00 | 1527768 | 6.416 | Alkaloid (Amine) | No known direct therapeutic benefit; a synthetic building block. | |
| 12 | 2-Piperidinecarboxylic acid | 849 | C6H11NO2 | 7.181 | 84.00 | 1527768 | 6.416 | Amino Acid Derivative | Basis for anticonvulsant and neuroprotective agents. | [35] |
| 13 | DL-Glutamic acid | 611 | C5H9NO4 | 7.181 | 84.00 | 1527768 | 6.416 | Amino Acid | Neurotransmitter supporting cognition, learning, and memory. | [28,36–38] |
| 14 | 1,3-Propanediol, 2-ethyl-2-(hydroxymethyl)- | 172578 | C6H14O3 | 8.313 | 57.00 | 789197 | 3.314 | Alcohol | No intrinsic therapeutic effect. | |
| 15 | Sucrose | 5988 | C12H22O11 | 8.313 | 57.00 | 789197 | 3.314 | Carbohydrate (disaccharide) | Energy source in medical nutrition, emergency hypoglycemia treatment. | [28] |
| 16 | Butoxyacetic acid | 41958 | C6H12O3 | 8.313 | 57.00 | 789197 | 3.314 | Ether acid | Antimicrobial | [39] |
| 17 | 2-Penten-1-ol, (Z)- | 5364919 | C5H10O | 8.313 | 57.00 | 789197 | 3.314 | Unsaturated Alcohol | Mild antimicrobial properties (very limited therapeutic relevance). | [28] |
| 18 | 1,5-Pentanediol | 8105 | C5H12O2 | 8.313 | 57.00 | 789197 | 3.314 | Alcohol | Acts as a moisture-retainer; promotes wound healing when used topically. | [40] |
| 19 | Isoamyl nitrite | 8053 | C5H11NO2 | 8.313 | 57.00 | 789197 | 3.314 | Organic Nitrite | Immediate vasodilation, rapidly relieving chest pain in angina patients. | [41] |
| 20 | Lethane | 8196 | C9H17NO2S | 8.313 | 57.00 | 789197 | 3.314 | Alkane | No therapeutic benefit | [42] |

*(Continued)*

| ID# | Name | Pubchem CID | Chemical Formula | Retention Time | m/z | Area | Conc. (ug/mL) | Chemical Class | Possible therapeutic use | Ref. |
|---|---|---|---|---|---|---|---|---|---|---|
| 21 | Cyclopentanol | 7298 | C5H10O | 8.313 | 57.00 | 789197 | 3.314 | Cycloalcohol | No direct therapeutic role; sometimes used in formulations. | [43] |
| 22 | Cystine | 67678 | C6H12N2O4S2 | 9.995 | 44.00 | 138773 | 0.583 | Amino Acid (disulfide-linked) | Antioxidant support, promotes detoxification and boosts immunity. | [44] |
| 23 | Tetrahydro-4H-pyran-4-ol | 74956 | C5H10O2 | 9.995 | 44.00 | 138773 | 0.583 | Alcohol derivative (cyclic) | Antimicrobial and insecticidal | [45] |
| 24 | DL-Cystine | 595 | C6H12N2O4S2 | 9.995 | 44.00 | 138773 | 0.583 | Amino Acid (disulfide-linked) | Same as Cystine — supports skin health and reduces oxidative stress, Anti-inflammatory, osteoarthritis and rheumatoid arthritis | [28,44] |
| 25 | 1,3-Dioxolane, 4-methyl- | 66119 | C4H8O2 | 9.995 | 44.00 | 138773 | 0.583 | Cyclic Ether | No direct therapeutic effect. | |
| 26 | Norpseudoephedrine | 162265 | C9H13NO | 9.995 | 44.00 | 138773 | 0.583 | Alkaloid (Amine) | Acts as a stimulant and appetite suppressant; was used for weight loss. | [28,46] |
| 27 | Allantoic acid | 203 | C4H8N4O4 | 9.995 | 44.00 | 138773 | 0.583 | Ureide (nitrogenous compound) | Byproduct; minimal therapeutic importance (research on wound healing). | [47] |
| 28 | Urea, butyl- | 11595 | C5H12N2O | 9.995 | 44.00 | 138773 | 0.583 | Urea derivative | No known therapeutic effect, but Keratolytic emollient (Urea) | [28] |
| 29 | (-)-Norephedrine | 26934 | C9H13NO | 9.995 | 44.00 | 138773 | 0.583 | Alkaloid (Amine) | Nasal decongestant, boosts alertness, suppresses appetite; has stimulant effects. | [28,46] |
| 30 | 1,2,3,4-Butanetetrol, [S-(R*,R*)]- | 539117 | C12H18O8 | 9.995 | 44.00 | 138773 | 0.583 | Sugar Alcohol | No direct therapeutic role | |
| 31 | Glutaraldehyde | 3485 | C5H8O2 | 9.995 | 44.00 | 138773 | 0.583 | Dialdehyde | Potent disinfectant; used to sterilize medical equipment. | [28] |
| 32 | D-erythro-Pentose, 2-deoxy- | 150629 | C5H11O7P | 9.995 | 44.00 | 138773 | 0.583 | Sugar (Deoxysugar) | Foundational for DNA; critical in gene therapy research. | [28] |
| 33 | Piperazine, 2-methyl- | 66057 | C5H12N2 | 9.995 | 44.00 | 138773 | 0.583 | Heterocyclic Amine | Intermediate for antipsychotic, anti-tuberculosis, and antihistamine drugs. | [28,48] |
| 34 | .+/-.-Tetrahydro-3-furanmethanol | 139980 | C5H10O2 | 9.995 | 44.00 | 138773 | 0.583 | Furan derivative (Alcohol) | No direct therapeutic benefit. | – |
| 35 | 1,2-Ethanediamine, N-(2-aminoethyl)- | 118594 | C6H17N3 | 9.995 | 44.00 | 138773 | 0.583 | Diamine | Chelating agent; may help in heavy metal detox therapy. | [28] |
| 36 | Cyclopropyl carbinol | 75644 | C4H8O | 9.995 | 44.00 | 138773 | 0.583 | Cycloalcohol | No direct benefit; synthetic intermediate. | – |
| 37 | 1,4-Butanediamine, N-(3-aminopropyl)- | 4453621 | C16H40N6 | 9.995 | 44.00 | 138773 | 0.583 | Polyamine | Polyamine analog; involved in cell proliferation research (cancer, aging). | – |
| 38 | 2-Pentanamine | 12246 | C5H13N | 11.458 | 44.00 | 13458 | 0.057 | Alkaloid (Amine) | Precursor for possible cardiovascular drugs. | – |
| 39 | dl-Alanine | 602 | C3H7NO2 | 11.458 | 44.00 | 13458 | 0.057 | Amino Acid | Essential for protein metabolism and tissue repair, antidiabetic and help in neurotransmission | [28,49] |
| 40 | Octodrine | 10982 | C8H19N | 11.458 | 44.00 | 13458 | 0.057 | Alkaloid (Amine) | Sympathomimetic, boosts physical performance; enhances focus and energy. | [28] |

*(Continued)*

| ID# | Name | Pubchem CID | Chemical Formula | Retention Time | m/z | Area | Conc. (ug/mL) | Chemical Class | Possible therapeutic use | Ref. |
|---|---|---|---|---|---|---|---|---|---|---|
| 41 | 1,2-Propanediamine | 6567 | C3H10N2 | 11.458 | 44.00 | 13458 | 0.057 | Diamine | Intermediate in production of anti-fungal and antiviral agents. | [28] |
| 42 | 2-Methylaminomethyl-1,3-dioxolane | 541754 | C5H11NO2 | 11.840 | 44.00 | 32103 | 0.135 | Heterocyclic Amine | No direct therapeutic benefit; synthetic tool. | |
| 43 | Ethanol, 2-(methylamino)- | 8016 | C3H9NO | 13.061 | 44.00 | 18368 | 0.077 | Amine Alcohol | Precursor to anti-asthmatic and antihypertensive drugs. | [28] |
| 44 | Cathine | 441457 | C9H13NO | 13.061 | 44.00 | 18368 | 0.077 | Alkaloid (Amine) | Mild stimulant, appetite suppressant; natural in *Catha edulis* (khat). | [28] |
| 45 | Norephedrine, (.+/-.)- | 26934 | C9H13NO | 13.061 | 44.00 | 18368 | 0.077 | Alkaloid (Amine) | CNS stimulant, appetite suppressant; sympathomimetic agent. | [28] |
| 46 | Benzeneethanamine, N-methyl- | 10012396 | C12H15N | 13.061 | 44.00 | 18368 | 0.077 | Aromatic Amine | Analog of amphetamine; stimulant effects | [28,50] |
| 47 | Cyclobutanol | 76218 | C4H8O | 13.061 | 44.00 | 18368 | 0.077 | Cycloalcohol | No strong therapeutic role; solvent properties. | – |
| 48 | sec-Butylamine | 24874 | C4H11N | 13.472 | 44.00 | 13820 | 0.058 | Amine | No direct therapeutic use. | – |
| 49 | Heptacosanoic acid, methyl ester | 41517 | C28H56O2 | 13.906 | 74.00 | 58869 | 0.247 | Fatty Acid and Ester | Skin conditioning agent; cosmetics (indirect health benefits). | [28] |
| 50 | Eicosanoic acid, methyl ester | 598337 | C21H40O3 | 13.906 | 74.00 | 58869 | 0.247 | Fatty Acid and Ester | May have anti-inflammatory and moisturizing effects. | [28] |
| 51 | Methyl tetradecanoate | 31284 | C15H30O2 | 13.906 | 74.00 | 58869 | 0.247 | Fatty Acid and Ester | Emollient for skin hydration. | [28] |
| 52 | Triacontanoic acid, methyl ester | 12400 | C31H62O2 | 13.906 | 74.00 | 58869 | 0.247 | Fatty Acid and Ester | Moisturizer in cosmetics. | [28] |
| 53 | 1-Propanol, 2-amino-, (.+/-.)- | 5126 | C3H9NO | 15.265 | 44.00 | 11397 | 0.048 | Amine Alcohol | Involved in drug synthesis; no direct benefit itself. | – |
| 54 | Formamide, N,N-dimethyl- | 6228 | C3H7NO | 15.265 | 44.00 | 11397 | 0.048 | Amide | Toxic solvent; no therapeutic benefit. | [28] |
| 55 | 9,12-Octadecadienoic acid, methyl ester, (E,E) | 8203 | C19H34O2 | 15.840 | 44.00 | 14228 | 0.060 | Fatty Acid Methyl Ester (linoleic acid derivative) | Omega-6 fatty acid ester; supports heart and skin health, Anticancer | [28] |
| 56 | 10-Undecyn-1-ol | 76015 | C11H20O | 15.840 | 44.00 | 14228 | 0.060 | Alkyne Alcohol | Investigated for antimicrobial and antifungal activities. | [28] |
| 57 | 9-Dodecyn-1-ol | 117011 | C12H22O | 15.840 | 44.00 | 14228 | 0.060 | Alkyne Alcohol | Investigated for antimicrobial and antifungal activities. | [28] |
| 58 | 2-Octynoic acid | 21872 | C8H12O2 | 15.840 | 44.00 | 14228 | 0.060 | Alkyne Carboxylic Acid | Antifungal potential; could inhibit hepatitis C infections | [51] |
| 59 | 4-Undecyne | 143690 | C11H20 | 15.840 | 44.00 | 14228 | 0.060 | Alkyne | Experimental research chemical. | |
| 60 | 3-Undecyne | 143689 | C11H20 | 15.840 | 44.00 | 14228 | 0.060 | Alkyne | Experimental; no clear therapeutic application yet. | |
| 61 | 2(3H)-Furanone, dihydro-4-hydroxy- | 5318286 | C5H6O3 | 16.165 | 44.00 | 15973 | 0.067 | Lactone | Antioxidant, anti-inflammatory agent (potentially protective for cells). | |
| 62 | Decanoic acid, methyl ester | 8050 | C11H22O2 | 16.165 | 44.00 | 15973 | 0.067 | Fatty Acid and Ester | Used in MCT-based therapies for epilepsy. | [28] |
| 63 | Octanoic acid, methyl ester | 8091 | C9H18O2 | 16.165 | 44.00 | 15973 | 0.067 | Fatty Acid and Ester | MCT oils; supports brain energy metabolism (Alzheimer's research). | [28] |

*(Continued)*

**Table 3.** (Continued)

| ID# | Name | Pubchem CID | Chemical Formula | Retention Time | m/z | Area | Conc. (ug/mL) | Chemical Class | Possible therapeutic use | Ref. |
|---|---|---|---|---|---|---|---|---|---|---|
| 64 | 3,3-Dimethylpiperidine | 70942 | C7H15N | 17.831 | 44.00 | 35231 | 0.148 | Alkaloid (Amine) | Precursor for therapeutic drug design for σ₁ receptor | [52] |
| 65 | 1-Hexanamine, N-hexyl- | 519999 | C12H27N | 17.831 | 44.00 | 35231 | 0.148 | Alkaloid (Amine) | Intermediate; no intrinsic therapeutic action. | |
| 66 | 9-Octadecenamide, (Z)- | 1930 | C18H35NO | 18.868 | 44.00 | 25622 | 0.108 | Fatty Amide (oleamide-like) | Promotes sleep; studied for Antioxidant, antimicrobial, anti-anxiety and anti-inflammatory effects. | [53,54] |
| 67 | 2-Furanmethanol, 5-ethenyltetrahydro-.alpha.,. | 6431475 | C10H18O2 | 18.868 | 44.00 | 25622 | 0.108 | Furan Alcohol | No therapeutic benefit directly; used in flavor/fragrance. | [28] |
| 68 | 2-Nonen-1-ol | 61896 | C9H18O | 18.868 | 44.00 | 25622 | 0.108 | Unsaturated Alcohol | Fragrance; no therapeutic benefit directly. | [28] |
| 69 | 2-Nonen-1-ol, (E)- | 5364941 | C9H18O | 18.868 | 44.00 | 25622 | 0.108 | Unsaturated Alcohol | Fragrance; no therapeutic benefit directly. | [28] |

2-Pentanamine, dl-Alanine, Octodrine, and others, have concentrations as low as 0.057 µg/mL. Some compounds, like 1-Propanol, 2-amino-, Formamide, N,N-dimethyl-, and others, have concentrations near 0.048–0.060 µg/ml. It appears that amino acids, amines, and alcohols tend to be present at moderate to high concentrations. Compounds like long-chain esters (e.g., Heptacosanoic acid, methyl ester, Triacontanoic acid, methyl ester) tend to have moderate concentrations as well. Many compounds with higher concentrations also have moderate retention times in the range of 6–8 minutes. This suggests they might be relatively simple to separate and quantify in the chromatography system. Compounds with lower concentrations are sometimes those with longer retention times (i.e., more complex interactions with the stationary phase of the chromatography column), like Octodrine or 2-amino-1-propanol.

In addition to retention time, concentration data is crucial for quantifying the amounts of different compounds presents in the sample. These data help to identify and confirm the presence of specific compounds, which is useful in fields like pharmacology, toxicology, biochemical analysis, or environmental testing. In industrial settings (such as pharmaceutical or chemical manufacturing), maintaining the correct concentrations of plant chemical compounds is critical, and this table can help with that monitoring. This table shows the quantitative distribution of various compounds in the methanolic extract. Both retention time and concentration offer a comprehensive picture that could be used for both the identification and quantification of chemical compounds.

## Chemical classification of the identified compounds

The compounds listed Table 3 lists a diverse range of chemicals, including Alkaloids and Amines: ~30% (amines, piperidines, piperazines); Amino Acids and Derivatives: ~10% (glutamine, glutamic acid, alanine, cystine); Alcohols: ~20% (pentanediols, cyclopentanol, sugar alcohols); Carbohydrates: ~3% (sucrose, deoxy-sugar); Fatty Acids and Esters: ~15% (octanoic, decanoic, eicosanoic acid methyl esters); Other Organics: lactones, ethers, aldehydes, amides, alkynes (Fig 4). There are reports that alkaloids are important for their antioxidant, anti-inflammatory properties, while flavonoids protect against oxidative stress, inflammation, and cardiovascular benefits. Carbohydrates are the main source of energy, and Amino acids are crucial for protein biosynthesis, muscle and metabolic regulation.

## Study of possible therapeutic benefits of the identified compounds

Compounds with different molecular structures and chemical properties possess diverse biological or chemical relevance, such as its use in pharmaceuticals, food, cosmetics, or as a reagent (Table 3). Compounds like Hexanal are described

as having antifungal properties, while L-Glutamine is recognized for its role as an amino acid. Some compounds, such as Isoamyl nitrite, have specific uses like being a vasodilator, and Sucrose is used as a sweetening agent and in treating certain health conditions. Other compounds are used for medical purposes, including as stimulants, neurotoxins, anticonvulsants, anti-inflammatory agents, and antioxidants. Examples include Pidolic acid, used in the treatment of neurological conditions, and Norephedrine, which is used for nasal congestion and obesity treatment. Many compounds have specific roles in the treatment or management of various conditions such as stimulant or psychoactive agent (e.g., Propanenitrile, 3-amino-2,3-di(hydroxymino)-), antifungal agents (e.g., Hexanal), antioxidants (e.g., Pidolic acid, 9-Octadecenamide, (Z)-), neurotransmitters and nutritional supplement (e.g., DL-Glutamic acid), sympathomimetics for nasal congestion, urinary incontinence, and obesity treatment (e.g., Norephedrine) (Table 3).

Some compounds are used in the manufacturing of industrial products such as Intermediate for pharmaceuticals, dyes, and other organic chemicals (e.g., Cyclopentanol), Lubricants and surfactants (e.g., Heptacosanoic acid, methyl ester), Emollient or skin-conditioning agents (e.g., Methyl tetradecanoate) (Table 3). Some compounds are primarily used as flavoring agents or in the formulation of food additives: Flavoring agent (e.g., 2-Penten-1-ol, (Z)-, 2-Nonen-1-ol), Food Preservatives and Food Coloring Agents (e.g., 9-Octadecenamide, (Z)-). This information is highly relevant in contexts like chemical research, pharmaceutical development, and industrial chemical synthesis. The data suggests that each compound plays a distinct and essential role, either as a product in pharmaceutical manufacturing or as a chemical intermediate for further synthesis.

## Study of structural properties of the selected compounds

The GC-MS analysis of beetroot extract revealed a diverse array of bioactive compounds with distinct structural properties (Fig 5). These include small cyclic amides such as 2-pyrrolidinone and pidolic acid, as well as amino acid derivatives including DL-proline 5-oxo-methyl ester, L-glutamine and DL-glutamic acid, sulfur-containing cystine, and nitrogenous compounds (piperazine, norpseudoephedrine, and benzeneethanamine) (Fig 5). Oxygenated heterocycles, including tetrahydro-4H-pyran-4-ol, 2-furanmethanol derivatives, and 2-methylaminomethyl-1,3-dioxolane, contribute hydroxyl and ether

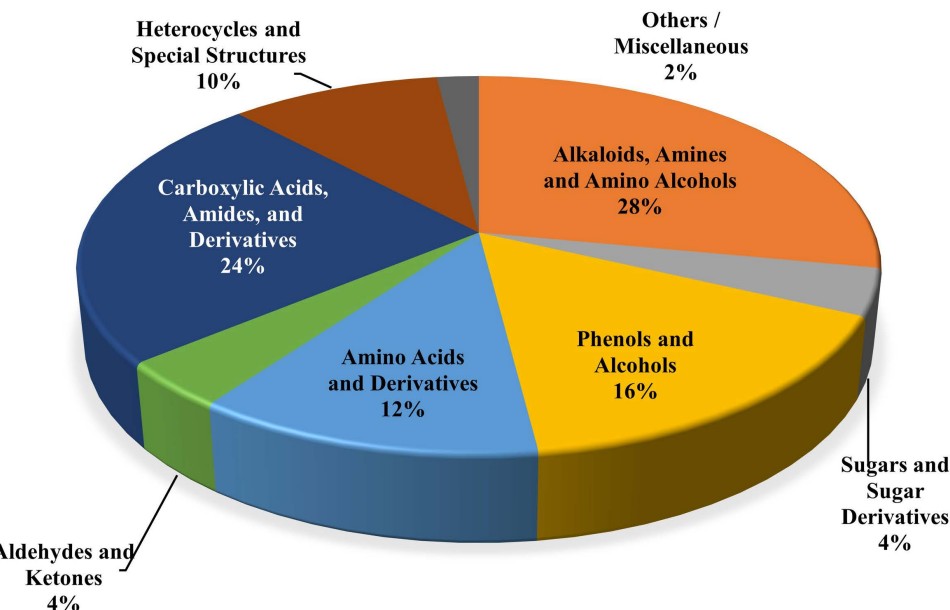

**Fig 4. Classification of the identified compounds based on chemical functional groups and structures in the methanolic extract of beetroot.**

functionalities that enhance polarity and reactivity. Lipophilic esters and amides such as methyl tetradecanoate, 9-octadecenamide, and butyl urea derivatives introduce hydrophobic moieties, which may influence membrane permeability and pharmacokinetic behavior. Additional functional groups, including nitrites (isoamyl nitrite), secondary amines, and carboxylic acids (butoxyacetic acid), diversify the chemical landscape of the extract, indicating a complex mixture of compounds capable of interacting with multiple molecular targets (Fig 5). Collectively, these structural features underscore the multifaceted chemical profile of beetroot, providing a foundation for its observed antioxidant, antihyperglycemic, and therapeutic potential.

## Estimation of molecular properties of the selected compounds

Analysis of molecular descriptors for each chemical compound, which can be used for understanding their chemical properties and predicting how they might behave in biological systems, chemical reactions, or physical environments. Compounds N-(3-aminopropyl)-1,4-Butanediamine and Heptacosanoic acid, methyl ester have a high number of rotatable bonds (17 and 26, respectively), indicating they are more flexible (Table 4 and S1 Table). Conversely, compounds like 3-Azabicyclo[3.2.2]nonane or Cyclopropyl carbinol have no rotatable bonds, meaning these compounds are rigid (Table 4). L-Glutamine, Sucrose, and DL-Proline, 5-oxo-, methyl ester are highly capable of both donating and accepting hydrogen bonds, which is characteristic of highly polar molecules (Table 4). On the other hand, Heptacosanoic acid, methyl ester and Triacontanoic acid, methyl ester have no hydrogen bond donors, making them non-polar and more likely to interact in non-polar environments like fats. The MR is a measure of a molecule's size and polarizability. Heptacosanoic acid, methyl ester (138 MR) and Triacontanoic acid, methyl ester (152.42 MR) have high values, indicating they are large, polarizable molecules, likely to interact with other large, non-polar substances (Table 4).

Smaller molecules N,N-dimethyl-formamide (MR = 20.02) have low MR, indicating they are smaller and less polarizable. Non-polar compounds like Heptacosanoic acid, methyl ester have low or zero TPSA, meaning they are lipophilic and can easily interact with lipid environments. Compounds like Heptacosanoic acid, methyl ester and Triacontanoic acid, methyl ester have very high Log P values (9.52 and 10.54, respectively), indicating they are hydrophobic and soluble in fat or oils (Table 4). Hydrophilic molecules like Sucrose and L-Glutamine have low Log P values (around -3.29 and -1.82), meaning they are water-soluble and not likely to interact with lipid-based environments (Table 4). L-Glutamine is highly polar and have low Log P values, making them likely to dissolve in water. Heptacosanoic acid, methyl ester and Triacontanoic acid, methyl ester have high Log P values, indicating they are more likely to interact with lipids and fats. Compounds like Heptacosanoic acid, methyl ester have many rotatable bonds, meaning they are more flexible and can adapt to different environments. Molecules such as Cyclopropyl carbinol have no rotatable bonds, making them rigid and less flexible. Compounds with more hydrogen bond donors and acceptors (e.g., L-Glutamine, Sucrose) tend to be more hydrophilic and interact more readily with water and other polar molecules. While compounds like Heptacosanoic acid, methyl ester lack hydrogen bond donors, which makes them more non-polar and lipophilic.

These molecular descriptors are important in drug development, biological research, and chemical synthesis. For example, Log P and TPSA help predict whether a compound can cross cell membranes, which is critical in drug design. The number of rotatable bonds and hydrogen bonding capacity can affect the compound's solubility, stability, and interactions with biological macromolecules. Table 4 provides a comprehensive set of descriptors for each chemical compound, offering insights into their physical, chemical, and biological behavior. These descriptors help predict the compound's solubility, polarity, ability to form interactions (such as hydrogen bonds), and potential biological activity.

## In silico Pharmacokinetics properties study of the selected compounds

Pharmacokinetic properties are critical for understanding how these compounds are absorbed, how they interact with biological systems, and their potential effects on various drug-metabolizing enzymes. Most compounds in the list have high GI absorption, which means they are likely to be absorbed well if taken orally (Table 5 and S2 Table). For example,

| Molecule Name | Structure | Molecule Name | Structure |
|---|---|---|---|
| 2-Pyrrolidinone, 5-(hydroxymethyl)- | | Norpseudoephedrine | |
| DL-Proline, 5-oxo-, methyl ester | | Urea, butyl- | |
| L-Glutamine | | Piperazine, 2-methyl- | |
| Pidolic acid | | Methyl tetradecanoate | |
| 2-Piperidinecarboxylic acid | | 9-Octadecenamide, (Z)- | |
| DL-Glutamic acid | | 2-Furanmethanol, 5-ethenyltetrahydro-.alpha. | |
| Isoamyl nitrite | | Benzeneethanamine, N-methyl- | |
| Cystine | | Butoxyacetic acid | |
| Tetrahydro-4H-pyran-4-ol | | 2-Methylaminomethyl-1,3-dioxolane | |

**Fig 5. 2D Structure of the selected molecules identified in the beetroot methanolic extract by GC-MS. Structures are drawn using ChemAxon's MarvinSketch software (MarvinSketch and Calculator Plugins, version 23.3.0).**

5-methyl-2-heptanamine, 3-amino-2,3-di(hydroxymino)- all show high absorption. However, some compounds like Sucrose, Cystine, and DL-Cystine have low GI absorption, suggesting they may be poorly absorbed from the GI tract. Several compounds such as 5-methyl-2-heptanamine, and Isoamyl nitrite are BBB permeant, meaning they have the

**Table 4. Physicochemical Properties analysis of the selected molecules identified by GC-MS.**

| Molecule Name | MW (g/mol) | Heavy atoms | Aromatic heavy atoms | Fraction Csp3 | Rotatable bonds | H-bond acceptors | H-bond donors | MR | TPSA (Å²) |
|---|---|---|---|---|---|---|---|---|---|
| 2-Pyrrolidinone, 5-(hydroxymethyl)- | 115.13 | 8 | 0 | 0.8 | 1 | 2 | 2 | 32.11 | 49.33 |
| DL-Proline, 5-oxo-, methyl ester | 143.14 | 10 | 0 | 0.67 | 2 | 3 | 1 | 37.04 | 55.4 |
| L-Glutamine | 146.14 | 10 | 0 | 0.6 | 4 | 4 | 3 | 33.54 | 106.41 |
| Pidolic acid | 129.11 | 9 | 0 | 0.6 | 1 | 3 | 2 | 32.72 | 66.4 |
| 2-Piperidinecarboxylic acid | 129.16 | 9 | 0 | 0.83 | 1 | 3 | 2 | 37.33 | 49.33 |
| DL-Glutamic acid | 147.13 | 10 | 0 | 0.6 | 4 | 5 | 3 | 32.4 | 100.62 |
| Isoamyl nitrite | 117.15 | 8 | 0 | 1 | 4 | 3 | 0 | 31.97 | 38.66 |
| Cystine | 240.3 | 14 | 0 | 0.67 | 7 | 6 | 4 | 55.1 | 177.24 |
| Tetrahydro-4H-pyran-4-ol | 102.13 | 7 | 0 | 1 | 0 | 2 | 1 | 26.28 | 29.46 |
| Norpseudoephedrine | 151.21 | 11 | 6 | 0.33 | 2 | 2 | 2 | 44.89 | 46.25 |
| Urea, butyl- | 116.16 | 8 | 0 | 0.8 | 4 | 1 | 2 | 32.25 | 55.12 |
| Piperazine, 2-methyl- | 100.16 | 7 | 0 | 1 | 0 | 2 | 2 | 37.47 | 24.06 |
| Methyl tetradecanoate | 242.4 | 17 | 0 | 0.93 | 13 | 2 | 0 | 75.5 | 26.3 |
| 9-Octadecenamide, (Z)- | 281.48 | 20 | 0 | 0.83 | 15 | 1 | 1 | 91.07 | 43.09 |
| Butoxyacetic acid | 151.21 | 11 | 6 | 0.33 | 2 | 2 | 2 | 44.89 | 46.25 |
| Benzeneethanamine, N-methyl- | 173.25 | 13 | 6 | 0.33 | 4 | 1 | 0 | 56.5 | 3.24 |
| 2-Methylaminomethyl-1,3-dioxolane | 125.21 | 9 | 0 | 1 | 0 | 1 | 1 | 43.06 | 12.03 |

ability to cross the BBB and potentially affect the central nervous system. L-Glutamine, Sucrose, and Cystine are not permeant to the BBB, which means they cannot easily enter the brain. Pgp is involved in pumping drugs out of cells, which can affect the drug's absorption and distribution (Table 5). Sucrose, Heptacosanoic acid, methyl ester, and Triacontanoic acid, methyl ester are Pgp substrates, suggesting they may be effluxed from cells, limiting their availability. The majority of compounds are not Pgp substrates (Table 5), which may enhance their retention within cells and improve bioavailability.

Some compounds act as inhibitors for various CYP enzymes, which are critical for drug metabolism. Eicosanoic acid, methyl ester inhibits CYP3A4 and CYP2C9, which are involved in the metabolism of many drugs, 9,12-Octadecadienoic acid, methyl ester, (E,E) inhibits CYP2C9 and CYP3A4, potentially interacting with drugs metabolized by these enzymes (Table 5). On the other hand, compounds like 5-methyl-2-Heptanamine show no inhibition of the CYP enzymes listed (Table 5), meaning they are unlikely to interfere with the metabolism of other drugs that are substrates for these enzymes. Compounds Heptacosanoic acid, methyl ester (log Kp = 0.28) and Triacontanoic acid, methyl ester (log Kp = 1.17) have poor permeability, suggesting they may have difficulty entering cells or crossing barriers like the BBB (Table 5). Compounds like 5-methyl-2-Heptanamine and 3-amino-2,3-di(hydroxymino)-propanenitrile with log Kp values around -5.46 have moderate permeability, indicating that they can more easily cross membranes.

Compounds that are BBB permeant (e.g., 5-methyl-2-heptanamine, Hexanal) might be useful for central nervous system (CNS) therapies. In contrast, non-BBB permeant compounds like L-Glutamine and Sucrose may be more relevant for peripheral effects (Table 5). The compounds that inhibit CYP enzymes (e.g., Eicosanoic acid, methyl ester for CYP3A4 and CYP2C9) could lead to drug interactions, potentially affecting the metabolism of co-administered drugs. This is particularly important when considering these compounds in therapeutic contexts. Compounds with low log Kp values (like Sucrose and Cystine) are less likely to cross cellular membranes easily. These compounds might require a different route of administration or may not be effective for drug delivery across certain biological barriers. Since most compounds have high GI absorption, they would likely be effective in oral formulations, except for those with poor permeability or low absorption like Sucrose, Cystine, and DL-Cystine (Table 5). Our data provides valuable insight into the pharmacokinetic profiles of various compounds, particularly regarding absorption, brain permeability, and drug interactions through CYP

**Table 5. Analysis of pharmacokinetics of the selected molecules identified by GC-MS.**

| Molecule Name | GI absorption | BBB permeant | Pgp substrate | CYP1A2 inhibition | CYP2C19 inhibition | CYP2C9 inhibition | CYP2D6 inhibition | CYP3A4 inhibition | log Kp (cm/s) |
|---|---|---|---|---|---|---|---|---|---|
| 2-Pyrrolidinone, 5-(hydroxymethyl)- | High | No | No | No | No | No | No | No | −7.82 |
| DL-Proline, 5-oxo-, methyl ester | High | No | No | No | No | No | No | No | −7.49 |
| Pidolic acid | High | No | No | No | No | No | No | No | −7.63 |
| 2-Piperidinecarboxylic acid | High | No | No | No | No | No | No | No | −8.73 |
| DL-Glutamic acid | High | No | No | No | No | No | No | No | −9.82 |
| Isoamyl nitrite | High | Yes | No | No | No | No | No | No | −5.79 |
| Cystine | Low | No | No | No | No | No | No | No | −11.37 |
| Tetrahydro-4H-pyran-4-ol | High | No | No | No | No | No | No | No | −6.96 |
| Norpseudoephedrine | High | Yes | No | No | No | No | No | No | −6.63 |
| Piperazine, 2-methyl- | Low | No | No | No | No | No | No | No | −7.22 |
| 9-Octadecenamide, (Z)- | High | Yes | No | Yes | No | Yes | No | No | −3.05 |
| Butoxyacetic acid | High | Yes | No | No | No | No | No | No | −6.63 |
| Benzeneethanamine, N-methyl- | Low | Yes | No | No | No | No | Yes | No | −5.7 |
| 2-Methylaminomethyl-1,3-dioxolane | High | No | No | No | No | No | No | No | −6.15 |
| dl-Alanine | High | No | No | No | No | No | No | No | −8.95 |

inhibition. These factors are crucial for assessing how these compounds might behave in the body, their potential for therapeutic use, and their likelihood of interacting with other drugs.

## Analysis of druglikeness of the selected compounds

Almost all compounds do not violate the Lipinski rules (0 violations), which suggests that they are likely to have reasonable oral bioavailability (Table 6 and S3 Table). However, compounds like Sucrose (2 violations) and Heptacosanoic acid, methyl ester (1 violation) show some potential for poor bioavailability. Sucrose (1 violation), L-Glutamine (3 violations), and Cyclopropyl carbinol (3 violations) have violations according to the Ghose filter, suggesting that these compounds may not fall within the optimal molecular ranges for drug-like molecules. Many compounds show 0 Veber violations, suggesting good oral bioavailability, but compounds like Cyclopropyl carbinol (3 violations) might face issues related to rotatable bonds and polar surface area. Similar to the Veber violations, many compounds show 0 Egan violations, indicating that they should have good permeability and absorption, while others, such as Cyclopropyl carbinol, show 3 violations, which could indicate challenges with absorption.

Most compounds have a bioavailability score of 0.55, suggesting they are likely to have moderate bioavailability (Table 6). However, Sucrose has a much lower score (0.17), indicating that it has poor oral bioavailability. Pidolic acid and 2-Octynoic acid have a higher score (0.85), indicating better bioavailability compared to most compounds (Table 6). Overall, our results present a useful analysis of drug-likeness for various compounds, based on physicochemical properties and their compliance with key drug discovery rules (Lipinski, Ghose, Veber, Egan, Muegge). Compounds with low violations (like Pidolic acid and 2-Octynoic acid) are more likely to have good oral bioavailability, while those with higher violations (like Sucrose and Cyclopropyl carbinol) may encounter challenges with bioavailability, absorption, and drug efficacy.

## Discussion

The phytochemical screening tests in our study confirm that Bangladeshi beetroot is rich in diverse phytoconstituents (e.g., alkaloids, flavonoids, polyphenols, tannins, terpenoids and carbohydrates) (Table 1). These results are in agreement with previous studies [2,8,9,55,56]. As an example, Rehman et al., (2021) investigated the biochemical profiling of

**Table 6. Analysis of Druglikeness of the selected molecules identified by GC-MS.**

| Molecule Name | Lipinski #violations | Ghose #violations | Veber #violations | Egan #violations | Muegge #violations | Bioavailability Score |
|---|---|---|---|---|---|---|
| 2-Pyrrolidinone, 5-(hydroxymethyl)- | 0 | 4 | 0 | 0 | 1 | 0.55 |
| DL-Proline, 5-oxo-, methyl ester | 0 | 4 | 0 | 0 | 1 | 0.55 |
| Pidolic acid | 0 | 4 | 0 | 0 | 1 | 0.85 |
| 2-Piperidinecarboxylic acid | 0 | 2 | 0 | 0 | 2 | 0.55 |
| DL-Glutamic acid | 0 | 4 | 0 | 0 | 2 | 0.56 |
| Isoamyl nitrite | 0 | 3 | 0 | 0 | 1 | 0.55 |
| Cystine | 0 | 1 | 1 | 1 | 2 | 0.55 |
| Tetrahydro-4H-pyran-4-ol | 0 | 3 | 0 | 0 | 1 | 0.55 |
| Urea, butyl- | 0 | 2 | 0 | 0 | 1 | 0.55 |
| Piperazine, 2-methyl- | 0 | 4 | 0 | 0 | 1 | 0.55 |
| Methyl tetradecanoate | 0 | 0 | 1 | 0 | 1 | 0.55 |
| 9-Octadecenamide, (Z)- | 1 | 0 | 1 | 0 | 1 | 0.55 |
| Butoxyacetic acid | 0 | 1 | 0 | 0 | 1 | 0.55 |
| Benzeneethanamine, N-methyl- | 0 | 0 | 0 | 0 | 2 | 0.55 |
| 2-Methylaminomethyl-1,3-dioxolane | 0 | 1 | 0 | 0 | 2 | 0.55 |

methanolic extracts of beetroot cultivated in Baluchistan and Sindh regions of Pakistan and observed the presence of various bioactive compounds, including phenols, flavonoids, alkaloids, saponins, and glycosides. Quantitative analysis of TPC and TFC across seven different solvents also revealed that Methanolic extracts yield higher levels of phenolics and flavonoids (Table 2). These results are consistent with the phytochemical fingerprint of methanolic beetroot extract cultivated in the Black Sea region [57]. The authors highlighted major phenolics and volatiles in the methanolic extracts [57].

The GC–MS analysis of the methanolic beetroot extract revealed a total of 69 phytoconstituents, distributed across several chemical classes, including alkaloids, amino acids and their derivatives, carbohydrates, fatty acids and their methyl esters, and organic compounds such as lactones, ethers, aldehydes and alkynes (Table 3 and Fig 4). Our GC-MS results are also aligned with the previous literature by Kusznierewicz B. *et al.* (2021, where the authors identified over 60 distinct phytochemicals employing advanced spectrometric platforms (HPLC-DAD, HPTLC, LC-Q-Orbitrap-HRMS) in beetroot cultivated in Poland, underscoring the complex regional phytochemical variation [58]. The GC-MS analysis of the methanolic extract of Egyptian beetroot led to the identification of only 17 compounds, including (Z,Z)-9,12-octadecadienoic acid; *n*-hexadecanoic acid, methyl ester; (Z,Z)-9,12-octadecadienoic acid; and (E)-9-octadecenoic acid [59]. Both our methanolic beetroot extract and Indian cultivars' beetroot showed linoleic acid and phytosterols, while Polish beetroot showed elevated betanin and ferulic acid levels [58]. Another study on American beetroot with GC-MS enabled the identification of 19 compounds, including six fatty acid methyl esters [7]. Thus, GC–MS analyses of beetroot samples from countries including Egypt, India, Poland and America have identified region-specific phytoconstituent patterns, with differences attributable to cultivar genotype, soil composition, climatic conditions, and post-harvest handling [7,57,59].

Our study identified Pyrrolidinone derivatives, 5-hydroxymethyl-2-pyrrolidinone, which have been studied as scaffolds for developing new drugs, including those with potential anticancer effects [60]. DL-Proline, L-Glutamine, DL-Cystine, Pidolic acid are common in protein biosynthesis or metabolic pathways. DL-Proline is commonly used in supplements for improving skin health, wound healing, and collagen production [61]. While L-Glutamine functions as a neurotransmitter and in nutritional supplements, it aids athletic to muscle recovery, reduces muscle soreness, and supports gut health [62]. Beyond protein biosynthesis, cystine plays a role in antioxidant defense as it is involved in the production of glutathione, a powerful antioxidant in the body [62]. There are reports that Pidolic acid exhibited effectiveness in a number of diseases,

including diabetes, Oxidative stresses, several neurological conditions, such as Alzheimer's disease and other cognitive disorders [28,63]. In dietary supplements, Pidolic acid can enhance cognitive function, but the evidence is limited. Although pidolic acid is included in some over-the-counter, non-prescription dietary supplements for the proposed purpose of facilitating cognitive or memory enhancement, most available research suggests exercising caution in their recommendation as much more research is necessary [28].

Like other amino acids, glutamine is biochemically important as a constituent of proteins. Glutamine can then be used as a nitrogen donor in the biosynthesis of many compounds, including other amino acids, purines, and pyrimidines. L-glutamine improves nicotinamide adenine dinucleotide (NAD) redox potential [64]. An oral formulation of L-glutamine was approved by the FDA in July 2017 for use in sickle cell disease [28,65]. Cystine is required for proper vitamin B6 utilization and is also helpful in the healing of burns and wounds, breaking down mucus deposits in illnesses such as bronchitis as well as cystic fibrosis [28,66]. Cysteine also assists in the supply of insulin to the pancreas, which is needed for the assimilation of sugars and starches [28,67].

Alanine is a non-essential amino acid occurs in high levels in its free state in plasma. It is involved in sugar and acid metabolism, increases immunity, and provides energy for muscle tissue, brain, and the central nervous system [68]. As a small molecule drug Alanine is under clinical trial phase III (across all indications) and has 3 investigational indications [69,70]. Alanine displays a cholesterol-reducing effect in animal model (NCI04) [68]. Thus L-glutamine, Cysteine and Alanine identified in the methanol extract of beetroot would provide an alternative source of these important amino acids.2-Piperidinecarboxylic acid also known as Pipecolic acid. Recently, Sato Y, et al (2024) reported that significantly increased level of endogenous Pipecolic acid in infected wheat plants with powdery mildew infection [71]. To date, there are no studies published addressing the biological effects of Pipecolic acid in animal or human. Cathine is an amphetamine that is propylbenzene substituted by a hydroxy group at position 1 and by an amino group at position 2 (the 1S,2S-stereoisomer). Cathine is also under clinical trial phase II and has 1 investigational indication, as a central nervous system stimulant and a psychotropic drug [28].

Although GC-MS is sensitive, its detection limits can be affected by factors like the sample preparation method and the type of detector used [72]. Some low-abundance compounds in plant extracts may not be detected or quantified accurately [72]. GC-MS primarily uses Electron Ionization (EI) for ionization, which can sometimes lead to fragmentation of the analytes, making it harder to identify or analyze larger molecules [72]. Plant extracts require extensive sample preparation, including extraction, purification, and possibly derivatization, which can be time-consuming and may lead to losses of volatile or unstable compounds [73]. Plant extracts are complex mixtures that often contain a wide variety of compounds, including essential oils, alkaloids, terpenoids, flavonoids, etc. This complexity can sometimes lead to co-elution of compounds during the GC separation, making it difficult to distinguish between closely related compounds [73]. Complementing GC-MS with other techniques, like LC-MS or direct infusion MS, can help overcome some of these limitations [7,12].

The in–silico analysis further supports the functional relevance of these compounds (Table 4 and 5). The flavonoids and phenolic acids comply with drug-likeness rules and have favorable predicted ADME profiles, suggesting they could be bioavailable when ingested. In contrast, glycine betaine and betanin violate Lipinski's criteria [15]; this data agrees with the fact that betanin is water-soluble and degraded in the gut, while betaine is highly polar. Fatty acids and long-chain amines also show poor solubility in water, limiting direct absorption (Tables 4 and 5). These predictions are in line with literature on these molecules: for example, beetroot glycine betaine is known more for osmoprotection and methyl-group donation than as a drug-like nutrient, and betanin's health effects are usually attributed to antioxidant activity in the gut rather than systemic action [15].

Finally, the identified compounds have notable nutraceutical and culinotherapeutic significance. The high antioxidant and anti-inflammatory potency of beetroot is largely ascribed to its betalains and polyphenols [2,4,55,56]. In practice, beetroot and its extracts are used in functional foods, sports drinks and dietary supplements to exploit effects like

blood-pressure reduction (via dietary nitrates) and oxidative stress mitigation. For instance, the ample inorganic nitrate and betalains observed in beetroot underlie its use in cardiovascular and liver-protective formulations [3,56]. Overall, our data corroborate beetroot's role as a "functional food" – the quantified phytochemicals can be harnessed in dietary therapies (culinotherapy) and nutraceutical product development [2–4,6,11,55,56,74].

## Conclusions

Overall, combined qualitative and quantitative analyses demonstrated the presence of a wide spectrum of phytoconstituents in beetroot cultivated in Bangladesh. GC-MS analysis of methanolic extract revealed the presence of bioactive compounds like amines, amino compounds, amino acids and derivatives, alcohols, polyols, carboxylic acids, esters, heterocyclic compounds, nitriles, isocyanates, and sugar-related compounds. Among the identified compounds, 5-(hydroxymethyl)-2-Pyrrolidinone, L-Glutamine, Methyl 5-oxo-L-prolinate, Pidolic acid, DL-Glutamic acid, Butoxyacetic acid, Isoamyl nitrite, Cystine, Norpseudoephedrine, 2-methyl-Piperazine, 3-Azabicyclo[3.2.2]nonane, 2-Pyrrolidinone, Piperazine, Octodrine and N-methyl-Benzeneethanamine are the major compounds that might contribute to biological activities such as antioxidant, anti-diabetic, antihypertensive, anti-inflammatory, anti-microbial and anti-cancer properties. Thus, the findings of the present study suggest that beetroot may serve as a potent source of bioactive compounds responsible for its pharmacological properties. Nevertheless, further research is required to isolate, characterize and purify the specific constituents responsible for its therapeutic effects

## Supporting information

**S1 Table. Analysis of lipophilicity of the selected 20 molecules identified by GC-MS.**
(DOCX)

**S2 Table. Analysis of water solubility of the selected 20 molecules identified by GC-MS.**
(DOCX)

**S3 Table. Analysis of medicinal chemistry of the selected 20 molecules identified by GC-MS.**
(DOCX)

## Author contributions

**Conceptualization:** Sarder Arifuzzaman, Zubair Khalid Labu.

**Data curation:** Sarder Arifuzzaman, Zubair Khalid Labu, Md. Mehedi Hasan, Abdullah Al Maruf, Sushanto Tappo.

**Formal analysis:** Sarder Arifuzzaman, Abdullah Al Maruf.

**Investigation:** Sarder Arifuzzaman, Zubair Khalid Labu, Md. Harun-Or-Rashid, Sushanto Tappo, Farhina Rahman Laboni.

**Methodology:** Sarder Arifuzzaman, Md. Mehedi Hasan, Sushanto Tappo.

**Project administration:** Zubair Khalid Labu, Md. Mehedi Hasan, Farhina Rahman Laboni.

**Resources:** Zubair Khalid Labu, Md. Harun-Or-Rashid, Md. Mehedi Hasan, Sushanto Tappo.

**Software:** Sarder Arifuzzaman, Md. Harun-Or-Rashid, Abdullah Al Maruf.

**Supervision:** Sarder Arifuzzaman.

**Validation:** Sarder Arifuzzaman, Farhina Rahman Laboni.

**Visualization:** Sarder Arifuzzaman.

**Writing – original draft:** Sarder Arifuzzaman.

**Writing – review & editing:** Sarder Arifuzzaman, Md. Harun-Or-Rashid.

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
