## [Decision Letter · Decision Letter 0]

6 Aug 2025

Response to Reviewers
Revised Manuscript with Track Changes
Manuscript
**Journal Requirements:**

1. Please provide a complete Data Availability Statement in the submission form, ensuring you include all necessary access information or a reason for why you are unable to make your data freely accessible. If your research concerns only data provided within your submission, please write "All data are in the manuscript and/or supporting information files" as your Data Availability Statement.

2. Some material included in your submission may be copyrighted. According to PLOS’s copyright policy, authors who use figures or other material (e.g., graphics, clipart, maps) from another author or copyright holder must demonstrate or obtain permission to publish this material under the Creative Commons Attribution 4.0 International (CC BY 4.0) License used by PLOS journals. Please closely review the details of PLOS’s copyright requirements here: PLOS Licenses and Copyright. If you need to request permissions from a copyright holder, you may use PLOS's Copyright Content Permission form.

Potential Copyright Issues: Figure 1: please (a) provide a direct link to the base layer of the map (i.e., the country or region border shape) and ensure this is also included in the figure legend; and (b) provide a link to the terms of use / license information for the base layer image or shapefile. We cannot publish proprietary or copyrighted maps (e.g. Google Maps, Mapquest) and the terms of use for your map base layer must be compatible with our CC-BY 4.0 license.

**Additional Editor Comments (if provided):**
**Reviewers' Comments:**

**Comments to the Author**

1. Does this manuscript meet PLOS Digital Health’s publication criteria?

Reviewer #1: Yes

Reviewer #2: Yes

2. Has the statistical analysis been performed appropriately and rigorously?

Reviewer #1: N/A

Reviewer #2: Yes

3. Have the authors made all data underlying the findings in their manuscript fully available (please refer to the Data Availability Statement at the start of the manuscript PDF file)?

Reviewer #1: Yes

Reviewer #2: No

4. Is the manuscript presented in an intelligible fashion and written in standard English?

Reviewer #1: No

Reviewer #2: Yes

Reviewer #1: These are the revoew comments to be address

The qualitative phytochemical screening is briefly mentioned, but specific details such as reagent names, test protocols, or controls used are not included. These should be described or referenced to ensure reproducibility.

The GC-MS method lacks critical information such as instrumentation parameters, retention times, column type, and identification approach (e.g., comparison with NIST library).

Reviewer #2: The manuscript is of interest because it explores bioactive compounds in a Beta vulgaris (beetroot) extract from Bangladesh and combines conventional phytochemical tests, gas chromatography–mass spectrometry (GC-MS) analysis, and in silico evaluations of pharmacokinetic properties. This multidisciplinary approach may help identify lead compounds for further drug development studies. The author also points out the lack of findings in phytochemical profiling of the beetroot grown in Bangladesh. The findings from this also add to the abundance of data in profiling beta vulgaris for phenotyping and comparisons with other origins from other geographical areas.

Strengths

1. Comprehensive Methodology Integration: The study integrates a multifaceted approach to not only profiling metabolite classes and identifying major constituents, but also performs drug-likeliness analysis, strengthening the findings.

2. Pharmacokinetic Modeling Transparency: The manuscript explicitly describes estimation formulas for key parameters (logBB, plasma protein binding (%PPB), half-life). For example, logBB=log(cerebral/plasma concentration) clarifies the physiological relevance of BBB permeability predictions.

3. GC-MS and In Silico Synergy: The combination of experimental chromatographic data with computational tools demonstrates practical applications in early drug discovery pipelines. Similar methodologies can be implemented for other use cases, especially given the highly reproducible description of the approach taken by the study team.

Potential Limitations

1. Ambiguous Formula Derivations: Equations like %PPB=dlogP+eMW+fHBA+g reference undefined coefficients (d,e,f,g). I presume they originate from the SwissADME models, and hence to the author, the models might be a black box and would limit access to coefficients/ weights. Additionally, Clarification on whether these models are generalized or empirically tailored to beetroot compounds. Include citations for original model derivations if applicable, might increase generalizability of the method.

2. Availability of data: The manuscript indicates “All data underlying the findings are fully available without restriction.” Please provide a precise statement regarding where and how the datasets can be accessed (e.g., repository name, accession numbers, DOIs) so that interested readers and reviewers can verify the results.

3. Software References: Chemaxon’s MarvinSketch and SwissADME tools are cited adequately (with URLs for Chemaxon), but versions or default parameters used lack specificity.

4. Comparison with other geographical areas: Expand discussion on elucidating the difference found by your identified compounds compare with those reported in other geographical or botanical studies.

Conclusion

This manuscript offers a robust computational framework integrating structural analysis with pharmacokinetic predictions. With revisions addressing these gaps, the study contributes a valuable methodology resource for early-stage drug discovery pipelines.

**Do you want your identity to be public for this peer review?** For information about this choice, including consent withdrawal, please see our Privacy Policy

Reviewer #1: **Yes: ** Dr.Sachi Nandan Mohanty

Reviewer #2: No

**Figure resubmission:****Reproducibility:** To enhance the reproducibility of your results, we recommend that authors of applicable studies deposit laboratory protocols in protocols.io, where a protocol can be assigned its own identifier (DOI) such that it can be cited independently in the future. Additionally, PLOS ONE offers an option to publish peer-reviewed clinical study protocols. Read more information on sharing protocols at https://plos.org/protocols?utm_medium=editorial-email&utm_source=authorletters&utm_campaign=protocols

---

## [Decision Letter · Decision Letter 1]

23 Sep 2025

Phytochemical Profiling and GC-MS Analysis of Bioactive Compounds in Methanolic Crude Extract of Beta vulgaris (BV) root from Bangladesh

PDIG-D-25-00287R1

Dear Dr. Arifuzzaman,

We're pleased to inform you that your manuscript has been judged scientifically suitable for publication and will be formally accepted for publication once it meets all outstanding technical requirements.

Within one week, you'll receive an e-mail detailing the required amendments. When these have been addressed, you'll receive a formal acceptance letter and your manuscript will be scheduled for publication.

An invoice for payment will follow shortly after the formal acceptance. To ensure an efficient process, please log into Editorial Manager at https://www.editorialmanager.com/pdig/ click the 'Update My Information' link at the top of the page, and double check that your user information is up-to-date. For billing related questions, please contact billing support at https://plos.my.site.com/s/.

Kind regards,

Hanieh Razzaghi

Section Editor

PLOS Digital Health

Additional Editor Comments (optional):

Reviewer #1:

Reviewer #2:

Reviewers' comments:

Reviewer's Responses to Questions

**Comments to the Author**

Reviewer #1: All comments have been addressed

Reviewer #2: All comments have been addressed

publication criteria?

Reviewer #1: Yes

Reviewer #2: Yes

3. Has the statistical analysis been performed appropriately and rigorously?

Reviewer #1: Yes

Reviewer #2: Yes

4. Have the authors made all data underlying the findings in their manuscript fully available (please refer to the Data Availability Statement at the start of the manuscript PDF file)?

Reviewer #1: Yes

Reviewer #2: No

5. Is the manuscript presented in an intelligible fashion and written in standard English?

PLOS Digital Health does not copyedit accepted manuscripts, so the language in submitted articles must be clear, correct, and unambiguous. Any typographical or grammatical errors should be corrected at revision, so please note any specific errors here.

Reviewer #1: Yes

Reviewer #2: Yes

Reviewer #1: All the comments well addressed in the reivesed manuscript

Reviewer #2: Thank you for making all the revisions from previous suggestions. I believe the manuscript now demonstrates acceptable statistical rigor and reproducibility in addressing the earlier suggestions. Additionally your edits to Phytochemical screenings and estimates for all solvents makes your methodology highly reproducible.

It is interesting to see overlapping findings from other geographical regions with some differences as well. And you have pointed out the differences attributable to cultivar genotype, soil composition, climatic conditions, and post-harvest handling. I am wondering if the method employed by other studies v/s yours have some implications on these differences in addition to the developmental and environmental factors.

Finally, I couldn't locate any datasets, or any files such as .mol or .sdf as you have mentioned in the manuscript. Will these files be made publicly available?

**Do you want your identity to be public for this peer review?** For information about this choice, including consent withdrawal, please see our Privacy Policy

Reviewer #1: Yes: Dr Sachi Nandan Mohanty

Reviewer #2: No
